# oMg: Opacity Matters in Material Modeling with Gaussian splatting

**Silong Yong**[*]   **Venkata Nagarjun Pudureddiyur Manivannan**[*]   **Bernhard Kerbl**
**Zifu Wan**        **Simon Stepputtis**        **Katia Sycara**        **Yaqi Xie**
Carnegie Mellon University
{silongy,vpudured,bkerbl,zifuw,sstepput,sycara,yaqix}@andrew.cmu.edu

## Abstract

Decomposing geometry, materials and lighting from a set of images, namely inverse rendering, has been a long-standing problem in computer vision and graphics. Recent advances in neural rendering enable photo-realistic and plausible inverse rendering results. The emergence of 3D Gaussian Splatting has boosted it to the next level by showing real-time rendering potentials. An intuitive finding is that the models used for inverse rendering do not take into account the dependency of opacity w.r.t. material properties, namely cross section, as suggested by optics. Therefore, we develop a novel approach that adds this dependency to the modeling itself. Inspired by radiative transfer, we augment the opacity term by introducing a neural network that takes as input material properties to provide modeling of cross section and a physically correct activation function. The gradients for material properties are therefore not only from color but also from opacity, facilitating a constraint for their optimization. Therefore, the proposed method incorporates more accurate physical properties compared to previous works. We implement our method into 3 different baselines that use Gaussian Splatting for inverse rendering and achieve significant improvements universally in terms of novel view synthesis and material modeling. Code is available at https://github.com/SilongYong/OMG.

## 1 Introduction

Inverse rendering, a long-standing problem in computer vision and graphics, aims to recover physical properties such as geometry, materials and lighting conditions from a set of images (Barron & Malik, 2013; Debevec, 2008). These properties are essential to downstream applications such as material editing, relighting and, from a broader perspective, real-to-sim transfer. However, it is well-known to be challenging and ill-posed because of the inherent ambiguity of the task: Different combinations of illumination, materials and geometry may lead to similar rendering results. Recent success in novel view synthesis (Mildenhall et al., 2021; Barron et al., 2021; Chen et al., 2022; Müller et al., 2022; Barron et al., 2023; 2022) inspired the use of implicit neural fields (NeRF-like methods) for inverse rendering (Zhang et al., 2021b; Yao et al., 2022; Boss et al., 2021; Jin et al., 2023; Srinivasan et al., 2021; Zhang et al., 2021a). On the other hand, 3D Gaussian Splatting (3DGS) (Kerbl et al., 2023) emerged as a recent, high-performance alternative to these slower architectures for novel view synthesis by introducing a much more compact scene representation and real-time rendering ability. Naturally, recent works combine 3DGS with inverse rendering (Liang et al., 2024; Gao et al., 2023; Jiang et al., 2024), taking advantage of its high-speed nature and compact geometry. By adding additional parameters that act as material properties for each Gaussian, they are able to provide physically plausible material modeling and novel view synthesis results using physically-based rendering (PBR) thanks to the explicit nature of 3DGS. While it is intuitive to take advantage of 3DGS for fast training and rendering with high quality, directly applying 3DGS for inverse rendering may result in sub-optimal modeling. Specifically, the correlation between the material attributes of each Gaussian (e.g., opacity, albedo, roughness, etc.)

---

[*]Joint collaborators in research inception

remains under-explored, leading to under-constrained modeling of the scene properties. Therefore, the inverse rendering results remain to be improved.

Observing the disentangled nature of 3DGS-based methods, our work, inspired by the Bouguer-Beer-Lambert law in radiative transfer, reveals the missing correlation between materials and opacity, two sets of parameters that are essential for physically-based rendering using 3D Gaussian Splatting. By doing so, our work introduces constraints at model level to align the model better with the physical world. Specifically, we propose to model opacity in 3DGS-based inverse rendering methods strictly following the form derived from the Bouguer-Beer-Lambert law, which states that the intensity of light should decrease exponentially when passing through an absorbing body. Intuitively speaking, the decrease rate depends not only on the path length that a light travels, but also on the material and number density of the absorbing body. An illustration of the law can be found in Fig. 1. Consider two different translucent materials glass and gas in Fig. 1. They react differently when hit by light. For instance, red light would be terminated by the gas while it's able to pass through the glass nearly unaffected as illustrated by Fig. 1. In other words, **opacity varies between different materials**. When designing the model, 3DGS-based inverse rendering methods (Liang et al., 2024; Jiang et al., 2024; Gao et al., 2023) overlook the fact that because of the disentangled representation, opacity is a standalone parameter that has no dependency on the material properties. As for NeRF-based methods that use alpha blending implicitly model the dependencies of opacity (volume density) and materials by using the same neural network to output different attributes. Therefore, for 3DGS-based methods, we propose to multiply the original opacity term with a material-dependent term and apply a physically correct activation function to give a better modeling of the physical property opacity itself. The material properties therefore serve as input for PBR as well as input for alpha-blending and receive gradients from both sides, facilitating a physically informed regularization to the model.

To verify the correctness and effectiveness of the proposed formulation, we analytically conduct Taylor expansion to our approach and compare the difference w.r.t. the original way of computing opacity. The takeaway message is that the original way is actually an approximation of our approach. Empirically, we apply the modification to 3 state of the art baselines, namely GaussianShader (Jiang et al., 2024), GS-IR (Liang et al., 2024) and R3DG (Gao et al., 2023) and conduct experiments on both synthetic and real-world data. The experimental results on Synthetic4Relight (Zhang et al., 2022), Shiny Blender (Verbin et al., 2022), Glossy Synthetic (Liu et al., 2023) and MIP-NeRF 360 (Barron et al., 2022) show that our approach enables universal performance improvement in terms of across different baselines and different data. The improve-

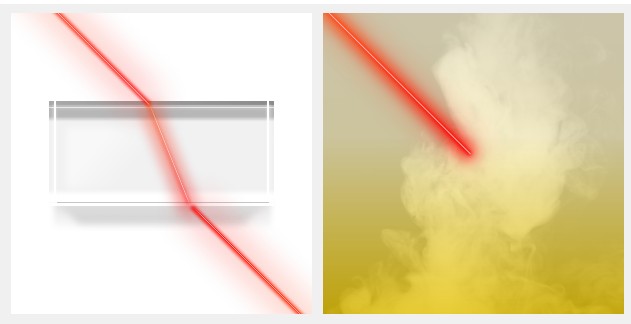

Figure 1: An illustration of the motivation that opacity depends on materials. Consider two translucent materials gas and glass. **Left**: a glass body that is hit by a light lets the light pass through with little reduction in intensity. **Right**: a gas that is hit by the same light make the light extinct inside it by absorbing it completely. This comparisons motivate our work. We perceive each Gaussian blob, paired with material properties, as an absorbing body that has its own absorption coefficient represented by opacity, which should consider material properties as influencing factors.

ments on material modeling, i.e., albedo estimation and roughness estimation, lead to better novel view synthesis and relighting results in terms of PSNR, SSIM and LPIPS. Our experimental results indicate the significance of incorporating physically correct priors and constraints into the model.

## 2 RELATED WORK

### 2.1 NEURAL SCENE REPRESENTATION

Recent trends in neural rendering (Mildenhall et al., 2021; Yu et al., 2021; Fridovich-Keil et al., 2022; Sitzmann et al., 2021; Yariv et al., 2021; Miller et al., 2024; Huang et al., 2024; Yong et al., 2024) have demonstrated impressive success in addressing visual computing problems like novel view synthesis. NeRF (Mildenhall et al., 2021) stands out as a representative by modeling the scene using an MLP to output volume density and color in a continuous space, which requires massive repeated queries for volume rendering during training and inference to synthesize images. Efforts have been made to improve the efficiency of NeRF by introducing additional data structures (Garbin et al., 2021; Müller et al., 2022; Hedman et al., 2021; Yu et al., 2021; Fridovich-Keil et al., 2022; Chen et al., 2022). However, the discretized nature of the data structures reduces the image quality to some extent. Recently, 3D Gaussian Splatting (3DGS) (Kerbl et al., 2023) comes out and is able to achieve fast rendering speed while maintaining the rendering quality, therefore drawing the community's interest. Plenty of works focus on improving 3DGS by reducing the memory footprint (Fan et al., 2023; Navaneet et al., 2023; Niedermayr et al., 2024), speeding up the training pipeline (Mallick et al., 2024; Höllein et al., 2024) and removing the heuristic designs for optimization (Kheradmand et al., 2024; Bulò et al., 2024). Another direction is on the application side, works have been done to apply 3DGS to different domains such as semantic understanding (Qin et al., 2024; Guo et al., 2024), time sequence modeling (Luiten et al., 2024; Lin et al., 2024) and robotic manipulation (Lu et al., 2024; Shorinwa et al., 2024), etc. The wide range of domains for application encouraged the focus of our work on inverse rendering techniques that use 3DGS as the backbone.

### 2.2 INVERSE RENDERING

Inverse rendering aims to decompose image observations into geometry, material and lighting conditions (i.e., scene properties) that support a myriad of downstream tasks, such as material editing and relighting (Li et al., 2023; Zhang et al., 2022; Kanamori & Endo, 2019; Yang et al., 2022). Normally, it is tackled by combining physically-based rendering with a differentiable renderer for optimization-based decomposition of scene properties (Kajiya, 1986; Chen et al., 2019). While being an inherently ambiguous problem, many works assume different constraints at input level, such as known lighting conditions (Bi et al., 2020; Srinivasan et al., 2021; Zhang et al., 2022), fixed lighting (Dong et al., 2014), unintended shadow (Verbin et al., 2024) or absence of shadow simulation (Zhang et al., 2021a). The combination of differentiable volume rendering and physically-based rendering has enabled modeling more complex conditions towards more realistic cases thanks to the emergence of neural scene representation (Zhang et al., 2021b; Yao et al., 2022; Liu et al., 2023; Verbin et al., 2022; Jin et al., 2023; Attal et al., 2024; Boss et al., 2021; Munkberg et al., 2022). Recently, 3DGS has revolutionized the field of neural scene representation. It is natural to expect inverse rendering could be tackled by 3DGS-based methods to incorporate fast and precise approximation of materials, geometry and lighting (Liang et al., 2024; Jiang et al., 2024; Gao et al., 2023). Despite the rapid success of providing plausible material estimation, the results remain to be improved, mainly because the constraints that a model should have for inverse rendering have been overlooked.

## 3 PRELIMINARIES

In this section, We will cover the basic concepts of 3D Gaussian Splatting, inverse rendering and the Bouguer-Beer-Lambert law as our work is built upon them.

### 3.1 3D GAUSSIAN SPLATTING

3D Gaussian Splatting (3DGS) (Kerbl et al., 2023) uses a set of points represented by 3D Gaussian primitives to model the 3D scene. Formally, each Gaussian $G_i$ is represented by a mean $\mu_i$, a covariance matrix $\Sigma_i$, an opacity $o_i$ and a view-dependent outgoing radiance $c_i$. Additionally, the covariance matrix is factorized into a quaternion $q_i$ and a scale $s_i$ and the outgoing radiance is treated as a set of spherical harmonics coefficients. For the inverse rendering problem, 3DGS-based methods (Liang et al., 2024; Jiang et al., 2024; Gao et al., 2023) assign material properties $m_i$ and

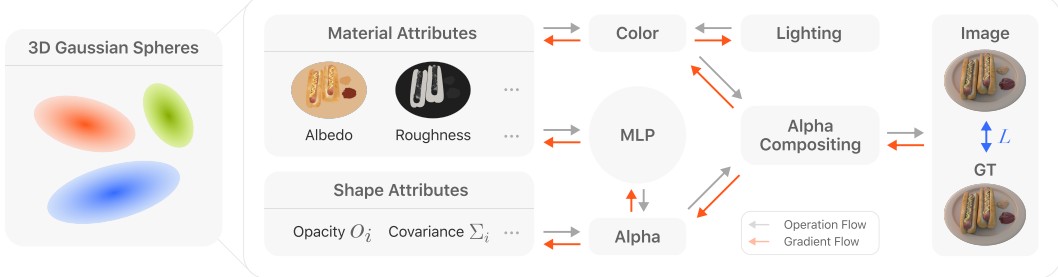

Figure 2: Pipeline overview. We add an additional block to 3DGS-based inverse rendering methods (Liang et al., 2024; Kerbl et al., 2023; Jiang et al., 2024; Gao et al., 2023). Specifically, instead of modeling opacity as a standalone parameter as done by previous works, we augment it against material. By introducing a neural network that takes material properties as input and output cross section and multiplying the opacity with it, we are able to incorporate the Bouguer-Beer-Lambert law into the model. During optimization, material properties not only receive the gradients from color through differentiable PBR, but also the gradients from alpha enforced by the neural network. By doing so, we add an additional constraint to material properties that makes the overall pipeline strictly follow the Bouguer-Beer-Lambert law.

normals $n_i$ to each Gaussian to support physically-based rendering. These sets of parameters are used to produce PBR color, which serves as $c_i$ in the original formulation.

After projecting the 3D Gaussians to 2D space using camera parameters $\theta$, 3DGS adopts $alpha$-blending for rendering the final color of a specific pixel location $x$ following

$$\hat{C}(x) = \sum_{i=1}^{N} c_i \alpha_i(x) T_i(x),$$

$$T_i(x) = \prod_{j=1}^{i-1} (1 - \alpha_j(x)),$$

$(1)$

where $N$ represents the sorted $N$ Gaussian points, $T_i$ is the accumulated transmittance of the first $i-1$ Gaussians and $\alpha_i(x)$ is computed using

$$\alpha_i(x) = o_i \exp(-\frac{1}{2}(x - R(\mu_i; \theta))^T R_\theta(\Sigma_i)^{-1}(x - R(\mu_i; \theta))), \qquad (2)$$

in which $R$ stands for the camera projection operation, $R_\theta$ is the operation of applying the camera projection to the covariance matrix and opacity $o_i$ is weighted by the probability determined by the distance of the pixel location to the mean of the projected Gaussian. Afterwards, color loss is computed between the rendered image and the ground truth:

$$L = ||C - \hat{C}||^2, \qquad (3)$$

## 3.2 THE RENDERING EQUATION

In inverse rendering, the outgoing radiance $C_o(x, \mathbf{v})$ at a specific surface point $x$ from viewing direction $\mathbf{v}$ is determined by the rendering equation (Kajiya, 1986):

$$C_o(x, \mathbf{v}) = \int_\Omega L_i(x, \mathbf{l}) f(\mathbf{l}, \mathbf{v}, x)(\mathbf{l} \cdot \mathbf{n}) d\mathbf{l}, \qquad (4)$$

where $L_i(x, \mathbf{l})$ is the incoming light from direction $\mathbf{l}$ at point $x$, $\mathbf{n}$ is the surface normal direction and $f$ is the Bidirectional Reflectance Distribution Function (BRDF). The integral is computed in the upper hemisphere of the surface.

## 3.3 THE BOUGUER-BEER-LAMBERT LAW

The Bouguer-Beer-Lambert law states the following (Mayerhöfer et al., 2020; Bouguer, 1729; Lambert, 1760; Beer, 1852): Consider a radiation of intensity $I$, i.e., a light of intensity $I$ passing through

an absorbing body, the change of its intensity is governed by the following differential equation:

$$dI_\nu = -\alpha_\nu I_\nu ds, \tag{5}$$

where subscript $\nu$ means that the term depends on the frequency of light and $\alpha_\nu$ is called the extinction coefficient, an attribute that determines how energy is dampened in the absorbing body.

Solving for intensity $I_\nu$, we get

$$I_\nu(s) = I_\nu(0)e^{-\alpha_\nu s}, \tag{6}$$

here $e^{-\alpha_\nu s}$ is the transmittance of region $s$, meaning that the light intensity is exponentially decreasing when traveling through the absorbing body. If the attenuation of light in the body doesn't contain scattering effects, the extinction coefficient can be computed as

$$\alpha_\nu = n\sigma_\nu, \tag{7}$$

where $n$ is the number density and $\sigma_\nu$ is cross section. These two terms often appear in chemistry, $n$ describes the degree of concentration of countable objects (usually particles) and $\sigma_\nu$ models the the probability that two particles will collide, which is determined by the types of the particles. An illustration of the Bouguer-Beer-Lambert law can be found in Fig. 3, the intensity of a light passing through a region filled with particles is reduced when "hit" with particles (normally the affecting area is bigger than the particle size). Intuitively speaking, the reduction in intensity is determined by the number of particles in the region that the light hits and the affecting area of the specific particle type, which corresponds to Eq. 7 and Eq. 6.

## 4 METHOD

We start by introducing the intuition behind our method. It is well-known that different materials react differently when hit by a beam of light, e.g., a wooden plank would diffuse the light and let nothing pass through, while a glass pane may allow light to travel inside with no dampening. In physics, this phenomenon is described by the Bouguer-Beer-Lambert law, suggesting that the opacity of an object depends on its material. Motivated by this, we do a series of derivation and propose a plug-and-play solution suggesting that opacity should be a function of material and the computation of transmittance should follow the exponential decay rate as in the Bouguer-Beer-Lambert law.

### 4.1 GAUSSIAN BLOBS AS ABSORBING BODY

Our approach arises from the Bouguer-Beer-Lambert law. Recall in Eq. 1, $T_i$ stands for the accumulated transmittance of the first $(i-1)$ Gaussians, indicating the reduction of light intensity of the $i$-th Gaussian when observed from the camera perspective. For each Gaussian $G_i$, the transmittance is calculated by $1 - \alpha_i(x)$. According to the Bouguer-Beer-Lambert law, the transmittance of an absorbing body should follow the form $e^{-\alpha_\nu s}$ or $e^{-n\sigma_\nu s}$ as in Eq. 6. Therefore we have

$$\begin{aligned} 1 - \alpha_i(x) &= e^{-n\sigma_\nu s}, \\ \alpha_i(x) &= 1 - e^{-n\sigma_\nu s}. \end{aligned} \tag{8}$$

A natural question then arises: **what is $n$, $\sigma_\nu$ and $s$ in the context of Gaussian Splatting?**

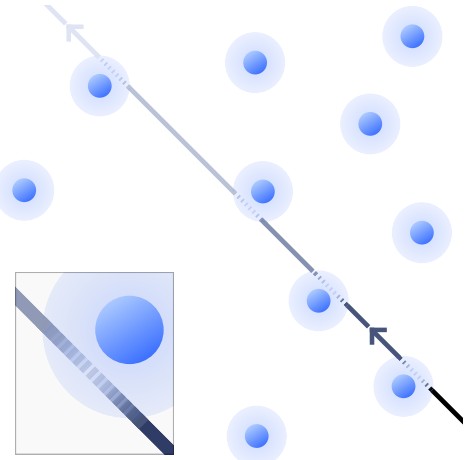

Figure 3: An illustration of the Bouguer-Beer-Lambert law. Consider a ray passing through an absorbing body consists of particles of some type. The cross section depends on the area that each particle would affect (usually bigger than the size of the particle and equals the size of the particle if treated as a hard sphere). When traveling, the intensity of the light would be reduced by the areas that each particle affected. The reduction in intensity is therefore affected by the area around the particles and the number of particles, corresponding to cross section $\sigma$ and number density $n$ in the main paper.

To answer this question, recall the definition of $\alpha_i(x)$ in Eq. 2. The term $o_i$ indicates how "dense" the Gaussian is, and the exponential term suggests that the Gaussian is densest at the center and becomes lighter when away from it as shown in Fig. 2. Since each Gaussian blob is not solid, it is natural to think of it as a blob of gas with its density following a Gaussian distribution. Notice that in this formulation, the density corresponds to the number density $n$ in Eq. 8. Formally, we have

$$n_i = o_i \exp(-\frac{1}{2}(x - R(\mu_i; \theta))^T R_\theta(\Sigma_i)^{-1}(x - R(\mu_i; \theta))). \tag{9}$$

As for path length $s$, since each Gaussian is "splatted" to a 2D plane, i.e. density is marginalized along the viewing axis, the distance of the light passing through each Gaussian can be set as a constant 1. In fact, the blending algorithm omits the depth of the Gaussians into the formulation of the opacity value at the pixel location. We therefore observed that we can treat this opacity value as the density and since there is no concept of depth on the 2D plane, each Gaussian is then treated equally from this perspective, therefore it is reasonable to treat the path length to constant 1.

## 4.2 MATERIAL NETWORK FOR CROSS SECTION MODELING

After deciding on the correspondence of existing terms, we found that there is still one term missing from the Bouguer-Beer-Lambert law, namely the cross section $\sigma_\nu$. Cross section, by definition, is determined by the type of particle that fills the region as illustrated in Fig. 3, namely the material of the region. On a macro level, cross section affects the color that one can perceive from the region. Since in inverse rendering, material properties, i.e., albedo, metalness and roughness, are explicitly included in the model (for 3DGS-based method they're set as additional parameters for each Gaussian), we can directly use the parameter for representing the cross section. In practice, a neural network $f(\cdot)$ that takes as input the material properties $m$ and outputs cross section $\sigma_\nu$ is introduced, namely

$$\sigma_\nu = f(m). \tag{10}$$

In summary, we have now determined the three terms suggested by the Bouguer-Beer-Lambert law in the context of Gaussian Splatting. Therefore, we can compute $\alpha_i(x)$ in Eq. 2 differently as

$$\begin{aligned} \alpha_i(x) &= 1 - e^{-n_i(x)\sigma_{\nu i}} \\ &= 1 - e^{-o_i G_i(x) f(m_i)}, \end{aligned} \tag{11}$$

where $o_i$ is the original opacity term for the $i$-th Gaussian, $G_i(x)$ is the normal distribution in Eq. 2, $f$ is the neural network that produces cross section and $m_i$ is the material property.

## 4.3 GRADIENT FLOW

Consider $c_i$ in Eq. 1, in inverse rendering: it is computed using the material properties in Eq. 4. Therefore, our formulation of material takes gradients not only from the color term itself, but also from the $\alpha$ term. In this case,

$$\frac{\partial L}{\partial m_i} = \frac{\partial c_i}{\partial m_i}\frac{\partial L}{\partial C}\alpha_i T_i + c_i\frac{\partial L}{\partial C}T_i\frac{\partial \alpha_i}{\partial m_i} - \sum_{j=i+1}^{N} c_j\frac{\partial L}{\partial C}T_j\alpha_j\frac{1}{1-\alpha_i}\frac{\partial \alpha_i}{\partial m_i}, \tag{12}$$

where $L$ stands for loss. Our formulation augments the gradient for material learning, allowing the gradient to flow from opacity to serve as an additional constraint in a physics-inspired way, as illustrated by Fig. 2.

## 4.4 ANALYSIS

Apart from the derivation of our formulation from the Bouguer-Beer-Lambert law, we found that our formulation could also be interpreted from two other perspectives, indicating the correctness of our approach.

**NeRF.** The alpha-blending algorithm in NeRF (Mildenhall et al., 2021) takes the form

$$C(r) = \sum_{i=1}^{N} c_i(1 - \exp(-\sigma_i\delta_i))T_i,$$

$$T_i = \exp(-\sum_{j=1}^{i-1} \sigma_j\delta_j), \tag{13}$$

in which $r$ stands for the ray direction, $c_i$ is a view-dependent color of the sampled points, $\sigma_i$ is volume density and $\delta_i$ is the distance between two sampled points. In this formulation, $\alpha_i = 1 - \exp(-\sigma_i\delta_i)$, which takes exactly the same form as our derivation from the Bouguer-Beer-Lambert law. Neural field-based methods (Bi et al., 2020; Boss et al., 2021) that use ray marching in a particle-based manner for inverse rendering handle cross section implicitly by using the same MLP for modeling volume density and material, suggesting the necessity of introducing cross section into the **disentangled** representation of 3DGS-based methods.

**Taylor Expansion.** Through Taylor expansion w.r.t. the function $\alpha(t) = 1 - \exp(-t)$, we have

$$\begin{aligned} \alpha(t) &= 1 - e^{-t} \\ &= 1 - (1 - t + o(t)) \\ &= t + o(t), \end{aligned} \tag{14}$$

where $o(t)$ is the little-o notation. This indicates that Eq. 2 is an approximation to the function $\alpha_i(x) = 1 - \exp(-o_i G_i(x))$, which is the exact same form as Eq. 11, omitting the newly introduced cross section $f(m_i)$. This derivation indicates the mathematical correctness of the newly introduced activation function.

## 5 EXPERIMENTS

We discuss the implementation details to show the experimental design and present experimental results on synthetic and real-world data to provide a comprehensive understanding of our method.

### 5.1 IMPLEMENTATION DETAILS

Since our proposed method is independent of the underlying model for inverse rendering, we can freely plug it into three state-of-the-art baselines, namely GaussianShader (Jiang et al., 2024), GS-IR (Liang et al., 2024) and R3DG (Gao et al., 2023) for evaluation. All the experiments are conducted on a single NVIDIA RTX 6000 Ada Generation GPU and we report the reproduced baseline results for fair comparison. We use an MLP that has 2 hidden layers with size 128 for the hidden dimension and ReLU as activation function for cross section and the output activation is a sigmoid function. The input to the MLP is designed as follows: for GaussianShader, an end-to-end trained material modeling pipeline, we directly input the albedo, roughness and specular tint terms as in their original model for MLP input. For GS-IR and R3DG, since they follow a two-stage training pipeline, in which the first stage uses normal rendering for geometry reconstruction and second stage uses physically-based rendering for material modeling, we input the material properties as well as the SH coefficients used for color in the first stage to the MLP. The logic behind this design is to make the optimization of the MLP in the first stage meaningful. We follow the experimental settings as provided by the original baselines. GaussianShader is evaluated on Shiny Blender (Verbin et al., 2022) and Glossy Synthetic (Liu et al., 2023) as in the orignal work. For R3DG, Synthetic4Relight dataset (Zhang et al., 2022) is used for getting material modeling evaluation results. GS-IR is evaluated on MIP-NeRF 360 dataset (Barron et al., 2022) to provide insights on real-world data. For albedo and relighting evaluation, we follow the standard practice to standardize the albedo prediction against the ground truth.

### 5.2 RESULTS ON SYNTHETIC DATA

**Material Modeling.** We first evaluate our proposed module on top of R3DG (Gao et al., 2023) for material modeling evaluation. As can be seen from Tab. 1, with the help of our modeling,

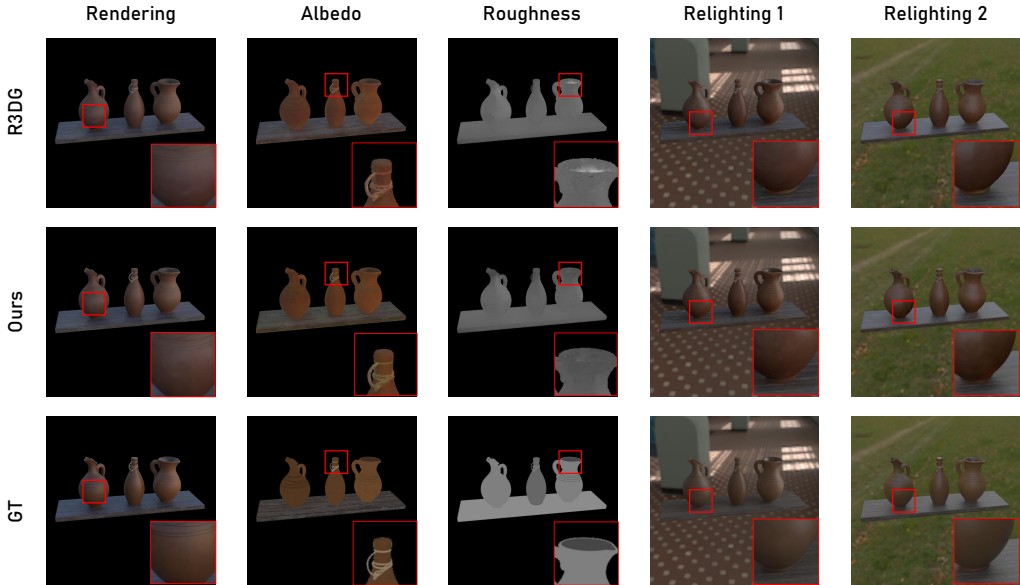

Figure 4: Qualitative comparison of albedo, relighting and roughness on Synthetic4Relight dataset (Zhang et al., 2022). Our method is able to decouple specular effects from albedo compared to R3DG, the baseline we implemented our method on. It is also capable of providing roughness estimation more precisely.We add a gamma correction to the relighting results to make the visual differences clearer.

Table 1: Quantitative results on Synthetic4Relight dataset (Zhang et al., 2022). Our method is able to provide significant improvements universally. The improvements in novel view synthesis and relighting can be attribute to the improved estimation of albedo and roughness thanks to the correct modeling we use for constraining material properties.

| | Novel View Synthesis | | | Relighting | | | Albedo | | | Roughness |
|---|---|---|---|---|---|---|---|---|---|---|
| | PSNR↑ | SSIM↑ | LPIPS↓ | PSNR↑ | SSIM↑ | LPIPS↓ | PSNR↑ | SSIM↑ | LPIPS↓ | MSE↓ |
| R3DG | 34.07 | 0.975 | 0.047 | 32.87 | 0.967 | 0.052 | 28.70 | 0.952 | 0.064 | 0.011 |
| R3DG + Ours | **34.47** | **0.977** | **0.046** | **33.21** | **0.969** | **0.050** | **29.31** | **0.954** | **0.059** | **0.007** |

R3DG is able to produce much better results universally. Our method outperforms the baseline on novel view synthesis by $0.4$ db PSNR, as well as on two other metrics. The relighting results shares similar performance boosts as novel view synthesis, which could be attributed to the significant improvements in albedo ($\sim 0.6$ db PSNR boost) and roughness estimation. Qualitatively speaking, as can be seen from the visualization in Fig. 4, our method is capable of producing albedo estimation with less specular effects and roughness estimation that aligns better with the ground truth, resulting in pleasing relighting results.

**Novel View Synthesis on Synthetic Data** Other than the results on Synthetic4Relight, we also implemented our method on top of GaussianShader (Jiang et al., 2024) to get more insights on how our model would perform on synthetic data. Tab. 2 showcases the result on Shiny Blender (Verbin et al., 2022) and Glossy Synthetic (Liu et al., 2023). Our method is able to produce significant improvement in terms of novel view synthesis when built on top of GaussianShader (specifically $\sim 0.3$ db PSNR boost on Shiny Blender and $\sim 0.4$ db boost on Glossy Synthetic). Fig. 5 showcases some qualitative comparisons of our model with the GaussianShader baseline. While we did not add any constraint to how the model predicts normals, our method is able to produce much more satisfactory normal estimation compared to the baseline as well as better modeling of light reflection.

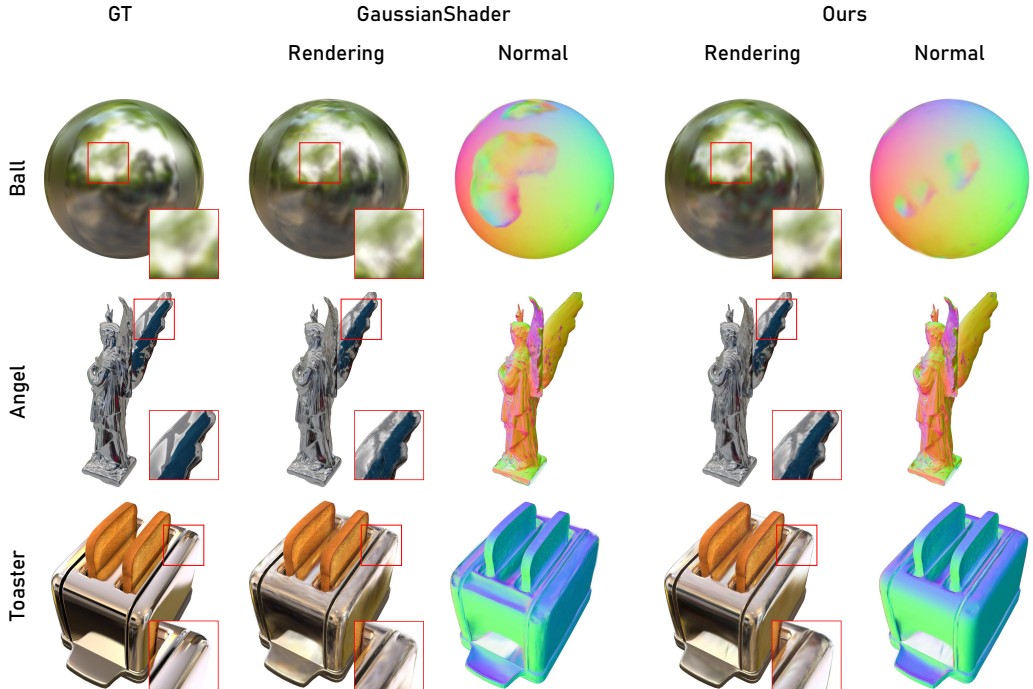

Figure 5: **Qualitative comparison on Shiny Blender (Verbin et al., 2022) and Glossy Synthetic (Liu et al., 2023) datasets.** Our method is able to provide satisfactory normal estimation compared to the baseline. Notice that our method does not include any additional supervision or constraints for normal estimation. Our method is also able to provide much more accurate view synthesis results when doing physically-based rendering. These results suggest the significance of introducing the physically correct model for inverse rendering.

Table 2: Quantitative results when combined with GaussianShader (Jiang et al., 2024). We compare with GaussianShader on Shiny Blender (Verbin et al., 2022) and Glossy Synthetic datasets (Liu et al., 2023). Our method is able to outperform GaussianShader on novel view synthesis in tems of all three standard image error metrics.

|  | Shiny Blender | | | Glossy Synthetic | | |
|---|---|---|---|---|---|---|
|  | PSNR↑ | SSIM↑ | LPIPS↓ | PSNR↑ | SSIM↑ | LPIPS↓ |
| GaussianShader | 30.77 | 0.953 | 0.081 | 27.25 | 0.925 | 0.083 |
| GaussianShader + Ours | **31.07** | **0.957** | **0.076** | **27.68** | **0.929** | **0.081** |

## 5.3 RESULTS ON REAL-WORLD DATA

Apart from the results on synthetic data, we also analyze the effect of our method when applied to real-world data. To verify the plug-and-play attribute of our method, we apply it to GS-IR (Liang et al., 2024) and conduct experiments on the MIP-NeRF 360 dataset (Barron et al., 2022). The experimental results shown in Tab. 3 indicate a significant universal improvement in terms of novel view synthesis thanks to the correct modeling and better estimation of materials as indicated by the results in Sec. 5.2. Qualitatively speaking, our method is able to produce less blurry images compared to the GS-IR baseline as shown by the bicycle scene in the first row in Fig. 6.

**Analysis.** Since our method augments opacity with a material-dependent term, it turns the opacity from a purely geometry-indicator to a combined representative of geometry and material. That being said, the optimization of geometry is also affected by our method. This is obvious if we consider the gradient of $\alpha_i$ w.r.t. $o_i$:

$$\frac{\partial \alpha_i}{\partial o_i} = G_i f(m_i)(1 - \alpha_i), \tag{15}$$

| Mip-NeRF 360 | | Avg. | Bicycle | Flowers | Garden | Stump | Treehill | Room | Counter | Kitchen | Bonsai |
|---|---|---|---|---|---|---|---|---|---|---|---|
| PSNR↑ | GS-IR | 25.38 | 23.31 | **20.43** | 25.76 | 25.46 | 21.75 | 28.32 | 26.31 | 28.66 | 28.44 |
| | Ours | **25.85** | 24.09 | 20.16 | **26.24** | **25.54** | **21.97** | **29.64** | **26.85** | **28.92** | **29.27** |
| SSIM↑ | GS-IR | 0.760 | 0.688 | **0.542** | 0.808 | 0.717 | 0.583 | 0.874 | 0.847 | 0.877 | 0.901 |
| | Ours | **0.767** | **0.720** | 0.510 | **0.824** | **0.720** | **0.586** | **0.897** | **0.859** | **0.884** | **0.905** |
| LPIPS↓ | GS-IR | 0.265 | 0.269 | **0.368** | 0.154 | 0.256 | **0.367** | 0.281 | 0.255 | 0.179 | 0.254 |
| | Ours | **0.254** | **0.230** | 0.401 | **0.140** | **0.245** | 0.368 | **0.247** | **0.244** | **0.168** | **0.243** |

Table 3: **Quantatitive Comparison on Mip-NeRF 360 (Barron et al., 2022).** The results show that our approach, when combined with GS-IR, can significantly improve the novel view synthesis results of real-world scenes on MIP-NeRF 360 (Barron et al., 2022). Thanks to the constraint we add to opacity, our method is able to produce $\sim 0.5$ db PSNR gains along with SSIM and LPIPS improvements.

pixel location $x$ is omitted for simplicity. We hypothesize that since MIP-NeRF 360 supports SfM initialization, it helps our method by easing the burden for geometry reconstruction, therefore leading to more improvements compared to synthetic data which uses random initialization.

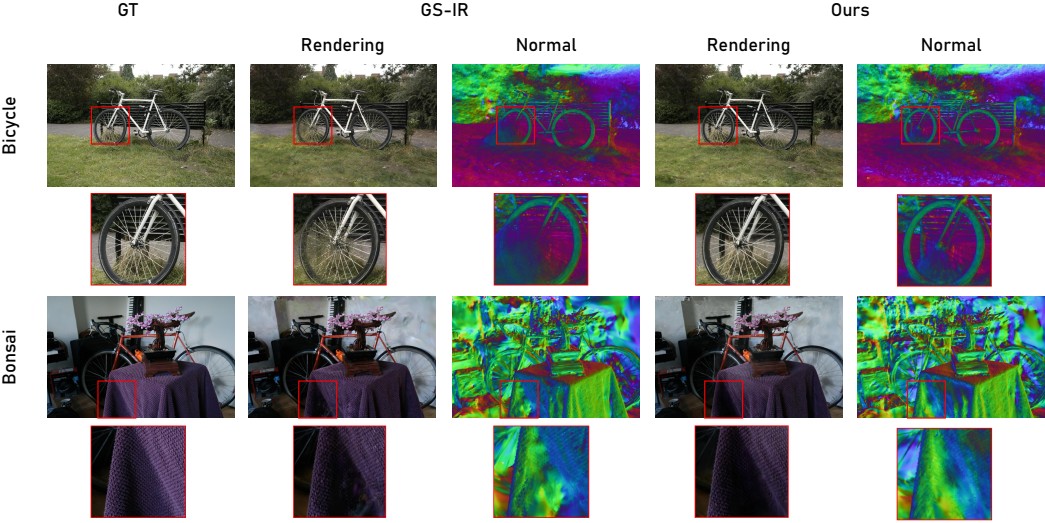

Figure 6: **Qualitative Comparison of GS-IR on MIP-NeRF 360 (Barron et al., 2022).** Our method is able to provide much more accurate novel view synthesis results compared to the baseline GS-IR (Liang et al., 2024) in which we plugged our model. The blurriness in the Bicycle scene in the first row predicted by GS-IR is overcome by our approach. The second row showcase that our approach is able to provide much more fine-grained and accurate normal estimation.

## 6 CONCLUSION

In this work, we propose OMG, a plug-and-play module for inverse rendering based on Gaussian Splatting. Inspired by the Bouguer-Beer-Lambert law, we derive the exact form of how opacity should be modeled in a particle-based model. We analyze the proposed formulation against NeRF-based methods and from a mathematical perspective, providing more insights in the right modeling for inverse rendering. Our method also reveals again the similarity between NeRF-based methods and 3D Gaussian Splatting. To validate the effectiveness and the plug-and-play nature of the proposed modeling, we implement it on top of three different state-of-the-art baselines and conduct comprehensive experiments on synthetic and real-world data. The significant improvements showcased by the experiments indicate the correctness of the modeling and point out the significance of the right prior and modeling for visual computing. We believe our work could inspire

the community to pay more attention to developing the physically correct model and introducing right priors for modeling the 3D world.

**Limitation.** In OMG, the dependency of cross section on the frequency of light is not included in the modeling. Future work could introduce the correct dependency of cross section on frequency of light to provide more accurate modeling of the physical world. As shown in Sec. D and Fig. D6, the learned cross section is still not ideal if applied to the specular region with strong lighting. This motivates us to explore more on how to regularize the learning of materials and cross section in a more precise way.

## ETHICS STATEMENT

Our work focuses on incorporating physically correct priors to improve inverse rendering results, aiming to bring new insights to visual computing community. Our research does not involve human subjects, sensitive data, or any practices that pose privacy or security concerns.

## REPRODUCIBILITY STATEMENT

The baselines used in our experiments, such as GaussianShader and GS-IR, are open-source and publicly available. We have detailed our experimental setup, and other design choices, in Section 5 of the main paper and Section A in the Appendix to ensure reproducibility. Please also be assured that the code implementation of this work will be made publicly accessible.

## ACKNOWLEDGEMENT

The author would like to thank Julia Du for the help of the title and the beautiful figures. This work has been funded in part by the Army Research Laboratory (ARL) award W911NF-23-2-0007 and W911QX-24-F-0049, DARPA award FA8750-23-2-1015, and ONR award N00014-23-1-2840.

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

# oMg: Opacity Matters in Material Modeling with Gaussian splatting

## Appendix

## A  Hyperparameters

The neural network introduced uses 0.001, 0.0001, 0.0005 and 0.007 as learning rate respectively for Synthetic4Relight, Shiny Blender, Glossy Synthetic and MIP-NeRF 360 datasets.The neural network is implemented as a fully-connected MLP with two hidden layers, 128 as hidden dimension and ReLU as activation function. We directly feed the material properties into the MLP without any encoding such as positional embedding.

## B  Full Derivation of Equations

Here we give the full derivation of Eq. 12.

$$
\begin{aligned}
\frac{\partial L}{\partial m_i} &= \frac{\partial L}{\partial C}\frac{\partial C}{\partial m_i} = \frac{\partial L}{\partial C}\sum_{j=1}^{N}\frac{\partial(c_j\alpha_j T_j)}{\partial m_i} \\
&= \frac{\partial L}{\partial C}\sum_{j=1}^{N}[\frac{c_j}{m_i}\alpha_j T_j + c_j\frac{\partial\alpha_j}{\partial m_i}T_j + c_j\alpha_j\frac{\partial T_j}{\partial m_i}] \\
&= \frac{\partial L}{\partial C}[\frac{\partial c_i}{\partial m_i}\alpha_i T_i + c_i T_i\frac{\partial\alpha_i}{\partial m_i} + \sum_{j=i+1}^{N}c_j\alpha_j\frac{\partial}{\partial m_i}\prod_{k=1}^{j-1}(1-\alpha_k)] \\
&= \frac{\partial L}{\partial C}[\frac{\partial c_i}{\partial m_i}\alpha_i T_i + c_i T_i\frac{\partial\alpha_i}{\partial m_i} + \sum_{j=i+1}^{N}c_j\alpha_j\frac{\partial}{\partial m_i}\exp(\sum_{k=1}^{j-1}(1-\alpha_k))] \\
&= \frac{\partial c_i}{\partial m_i}\frac{\partial L}{\partial C}\alpha_i T_i + c_i\frac{\partial L}{\partial C}T_i\frac{\partial\alpha_i}{\partial m_i} - \sum_{j=i+1}^{N}c_j\frac{\partial L}{\partial C}T_j\alpha_j\frac{1}{1-\alpha_i}\frac{\partial\alpha_i}{\partial m_i},
\end{aligned}
\tag{16}
$$

## C  Additional Results

We present the full results on Synthetic4Relight (Zhang et al., 2022), Shiny Blender (Verbin et al., 2022) and Glossy Synthetic (Liu et al., 2023). All the baseline results are reproduced by us as in the main paper. More visualization results can be found in Fig. C1, Fig. C2, Fig. C3, Fig. C4 and Fig. C5.

## D  Visualization of Cross Section

To understand what the cross section network is learning, we visualize some examples in Fig. D6. In the figure, the darker region indicates a lower cross section, meaning that it lets more light pass through whereas the brighter regions stands for the regions that have a higher cross section. The example shows that the windows as well as the plastic bag on the floor actually have a lower cross section assigned to them, which corresponds to human intuition.

| Scene | Method | Novel View Synthesis | | | Albedo | | | Relight | | | Roughness |
|---|---|---|---|---|---|---|---|---|---|---|---|
| | | PSNR ↑ | SSIM ↑ | LPIPS ↓ | PSNR ↑ | SSIM ↑ | LPIPS ↓ | PSNR ↑ | SSIM ↑ | LPIPS ↓ | MAE ↓ |
| Balloons | R3DG | 31.90 | 0.967 | **0.072** | 24.82 | 0.923 | 0.070 | 31.38 | 0.965 | 0.078 | **0.0009** |
| | Ours | **32.11** | **0.968** | 0.072 | **25.84** | **0.926** | **0.055** | **32.50** | **0.967** | **0.071** | 0.0011 |
| Hotdog | R3DG | 32.61 | 0.972 | 0.050 | 26.53 | 0.956 | 0.089 | **29.04** | 0.954 | 0.067 | 0.0357 |
| | Ours | **33.59** | **0.975** | **0.047** | **26.95** | **0.957** | **0.085** | 28.73 | **0.955** | **0.066** | **0.0217** |
| Chair | R3DG | 34.72 | 0.978 | **0.039** | 30.55 | 0.963 | 0.046 | 33.07 | 0.966 | **0.041** | **0.0036** |
| | Ours | **35.00** | **0.979** | 0.040 | **30.82** | **0.964** | **0.045** | **33.46** | **0.968** | **0.041** | 0.0039 |
| Jugs | R3DG | 37.03 | **0.984** | **0.027** | 32.88 | 0.967 | 0.064 | 37.98 | **0.984** | **0.023** | **0.0032** |
| | Ours | **37.17** | **0.984** | **0.027** | **33.64** | **0.968** | **0.049** | **38.16** | **0.984** | **0.023** | 0.0074 |

Table C1: Per-scene results on Synthetic4Relight dataset. For albedo reconstruction results, we follow the standard practice (Zhang et al., 2021b; Gao et al., 2023) and scale each RGB channel by a global scalar.

| **Shiny Blender** | | Avg. | Ball | Car | Coffee | Helmet | Teapot | Toaster |
|---|---|---|---|---|---|---|---|---|
| PSNR↑ | GaussianShader | 30.77 | 29.05 | **28.51** | **31.55** | **28.91** | **43.57** | 23.05 |
| | Ours | **31.07** | **29.65** | 28.14 | 31.29 | 28.79 | 43.48 | **25.07** |
| SSIM↑ | GaussianShader | 0.953 | 0.955 | **0.940** | **0.970** | **0.956** | **0.996** | 0.902 |
| | Ours | **0.957** | **0.961** | 0.939 | 0.969 | 0.955 | **0.996** | **0.924** |
| LPIPS↓ | GaussianShader | 0.081 | 0.146 | **0.047** | **0.085** | 0.086 | **0.011** | 0.110 |
| | Ours | **0.076** | **0.137** | 0.050 | 0.086 | **0.085** | **0.011** | **0.089** |

Table C2: Per-scene results on Shiny Blender dataset.

| **Glossy Synthetic** | | Avg. | Angel | Bell | Cat | Horse | Luyu | Potion | Tbell | Teapot |
|---|---|---|---|---|---|---|---|---|---|---|
| PSNR↑ | GaussianShader | 27.25 | 26.92 | 29.73 | **31.43** | 26.38 | 27.31 | **29.54** | 23.41 | 23.25 |
| | Ours | **27.68** | **28.24** | **29.91** | 31.00 | **26.62** | **27.32** | 29.40 | **24.67** | **24.31** |
| SSIM↑ | GaussianShader | 0.925 | 0.923 | 0.939 | **0.961** | **0.934** | **0.917** | **0.936** | 0.897 | 0.895 |
| | Ours | **0.929** | **0.929** | **0.940** | 0.959 | **0.934** | 0.916 | 0.935 | **0.907** | **0.909** |
| LPIPS↓ | GaussianShader | 0.083 | 0.071 | 0.085 | **0.058** | **0.056** | **0.066** | **0.096** | 0.136 | 0.098 |
| | Ours | **0.081** | **0.066** | **0.084** | 0.059 | 0.057 | 0.067 | 0.097 | **0.125** | **0.091** |

Table C3: Per-scene results on Glossy Synthetic dataset.

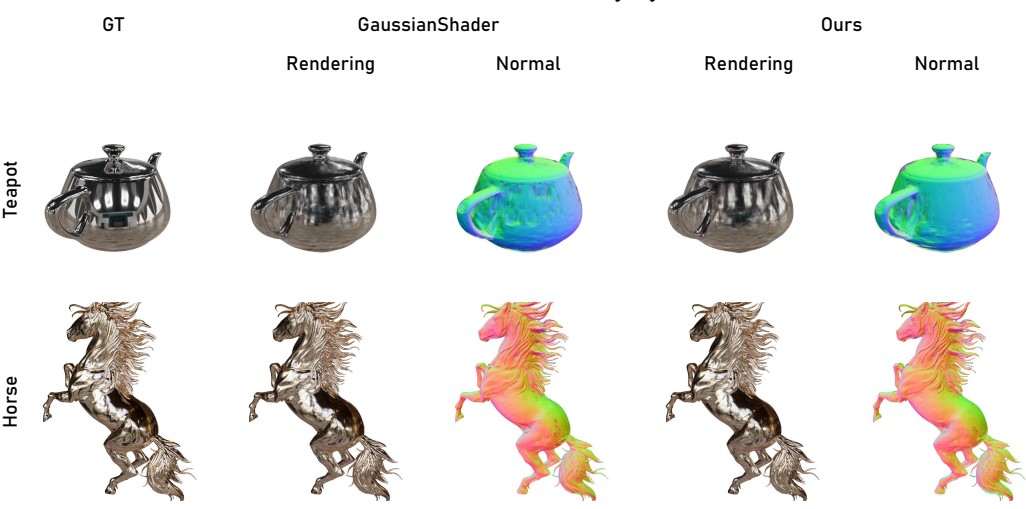

Figure C1: More qualitative example in comparison to GaussianShader on Glossy Synthetic dataset.

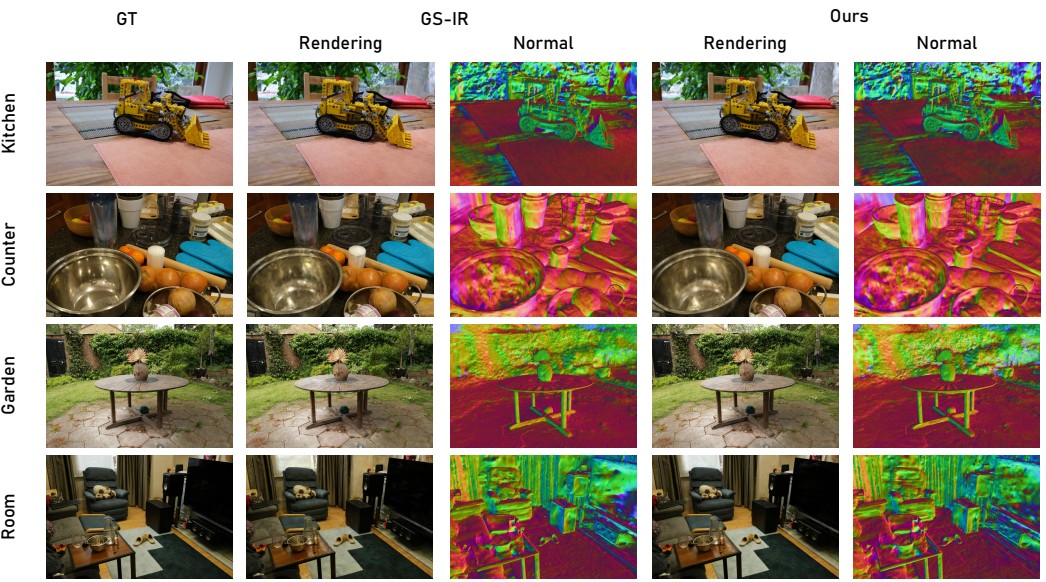

Figure C2: More qualitative example in comparison to GS-IR on MIP-NeRF 360 dataset.

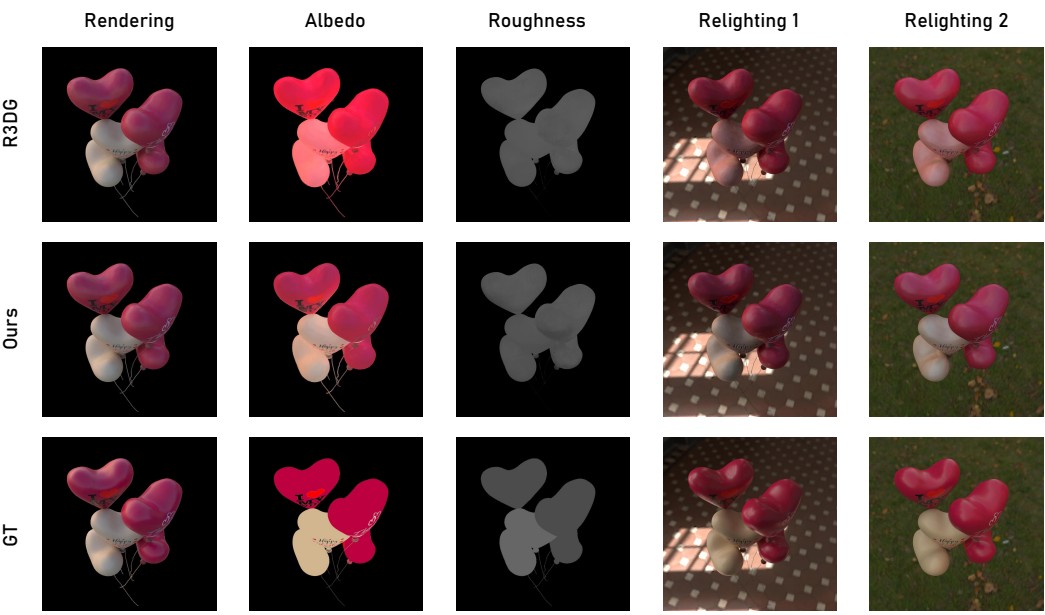

Figure C3: More qualitative example in comparison to R3DG on Synthetic4Relight dataset.

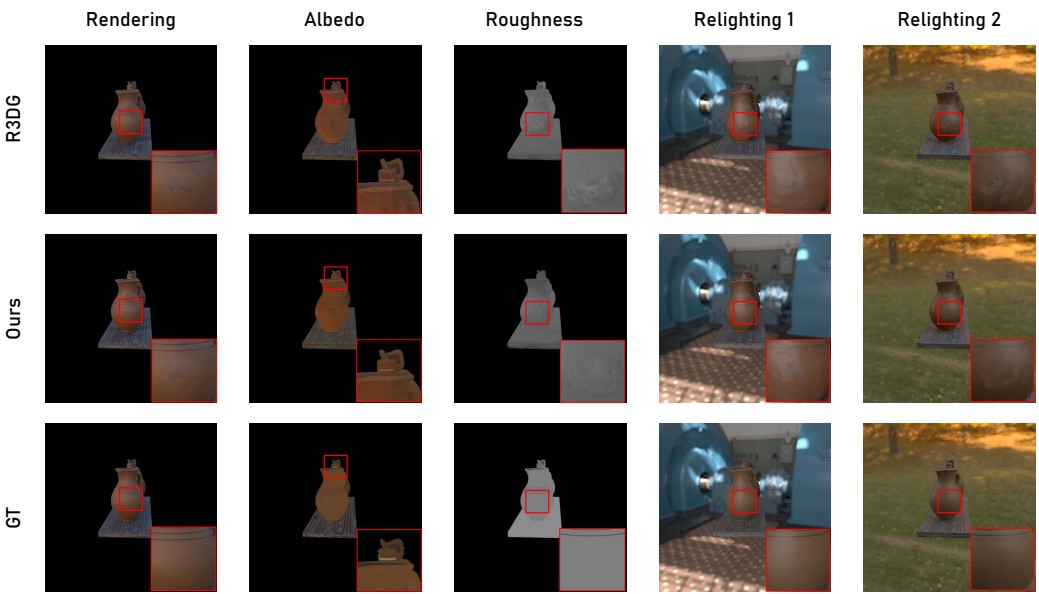

Figure C4: More qualitative example in comparison to R3DG on Synthetic4Relight dataset. We add a gamma correction to the relighting results to make the visual differences clearer.

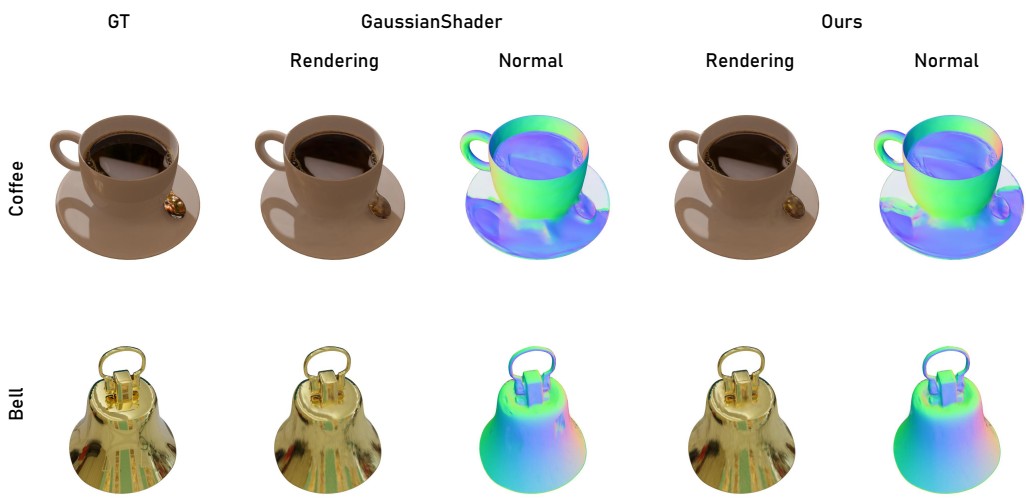

Figure C5: More qualitative example in comparison to GaussianShader on Glossy Synthetic and Shiny Blender datasets.

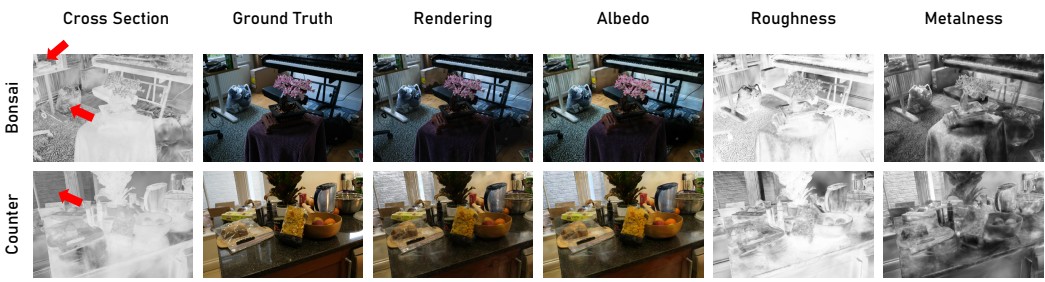

Figure D6: Visulization of the learned cross section on Mip-NeRF 360 dataset. Dark indicates the lower cross section (meaning it lets more light pass through) and bright stands for the higher cross section. The example shows that the windows as well as the plastic bag on the floor actually have a lower cross section assigned to them, which corresponds to human intuition.

