# OpenReview forum: "OMG: Opacity Matters in Material Modeling with Gaussian Splatting"
_ICLR.cc/2025/Conference — ICLR 2025 Poster_

### Official Review · Reviewer_RDFN · 2024-10-21

**Soundness:** 2
**Presentation:** 2
**Contribution:** 2
**Rating:** 6
**Confidence:** 3

**Summary:**

This manuscript introduces a 3D Gaussian Splatting (3D-GS) based inverse rendering task, where opacity is conditioned on material properties as suggested by the Bouguer-Beer-Lambert law. Comparisons within 3D-GS based inverse rendering tasks and ablation studies validate the effectiveness of this approach.

**Strengths:**

1. The Bouguer-Beer-Lambert law is interesting and technically sound.
2. Experiments validate the effectiveness of the proposed cross-section dependency.

**Weaknesses:**

1. Minor improvements have been observed compared to the baseline.
2. A comparison with NeRF-based inverse rendering is missing.
3. The claim "The emergence of 3D Gaussian Splatting has boosted it to the next level" in Line 15 is not convincing. GS-based inverse rendering has proven to be far less effective than SDF-based inverse rendering frameworks like NeRO[1] in terms of performance.
4. The case presented in Figure 1 considers translucent objects, where opacity is influenced by materials. However, the experiments conducted use opaque objects.


[1]Liu Y, Wang P, Lin C, et al. Nero: Neural geometry and brdf reconstruction of reflective objects from multiview images[J]. ACM Transactions on Graphics (TOG), 2023, 42(4): 1-22.

**Questions:**

1. Maybe experiments on transparent or translucent objects can support the theory.

---

> ### Author Response · Authors · 2024-11-21
> **Response to Reviewer RDFN**
>
> We’re thankful for your time and the valuable insights you’ve shared. Your input has significantly advanced our project. In response to your feedback, we proposed a brand new modeling for opacity in 3DGS-based inverse rendering based on the Bouguer-Beer-Lambert law and validated its effectiveness . We’ll address your concerns below.
>
> > 1. Minor improvements have been observed compared to the baseline.
>
> Thank you for your comment! We’d like to argue that the improvements are not minor. Our method outperformed baseline methods by ~0.4 PSNR, ~0.3 PSNR, ~0.4PSNR and ~0.5 PSNR on Synthetic4Relight, ShinyBlender, GlossySynthetic and Mip-NeRF 360 datasets respectively. Since PSNR is the log of MSE with base 10, our method is actually 2-3 times better than the baseline in terms of MSE. This is a common improvement scale in the relevant literatures[1, 2, 3]
>
> > 2. A comparison with NeRF-based inverse rendering is missing.
>
> Thank you for pointing this out! Since the baselines we build upon have already been compared with NeRF-based inverse rendering methods and generally outperform them, and our method is designed to be a plug-in module for 3DGS-based approaches, we focus on comparing our method against these baselines. While our approach could be integrated into NeRF-based inverse rendering, doing so would require significant effort to adapt and redesign the pipeline. We therefore consider this an exciting direction for future work.
>
> > 3. The claim "The emergence of 3D Gaussian Splatting has boosted it to the next level" in Line 15 is not convincing.
>
> Thank you for your comment! We’ve modified the manuscript in Ln. 15 to make it less ambitious. The intuition behind this sentence is that 3DGS-based methods introduce faster rendering speed, showing more potential than NeRF-based methods in terms of real-time rendering.
>
> >4. Maybe experiments on transparent or translucent objects can support the theory.
>
> Thank you for carefully investigating our manuscript! We’ve conducted experiments on translucent objects and showcased the results here. Since there are no existing baselines that do inverse rendering for translucent objects, we run the baseline that we use, namely GaussianShader, on the Dex-NeRF dataset as well as GaussianShader + our method. Since Dex-NeRF dataset doesn't provide train-test split, we splited the image set and used $\frac{2}{3}$ for training and rest for testing. We also run the original 3DGS on this dataset to showcase the upper bound for an inverse rendering method. The discovery is that our method still makes the baseline superior compared to the original version despite that both methods couldn't provide convicing results. The reason is that albedo, roughness and metalness are not the parameters for describing translucent objects. We argue that developing such a method is out of the scope of this work.
>
> | Method                | PSNR  | SSIM | LPIPS |
> | --------------------- | ----- | ---- | ----- |
> | 3DGS                  | 17.47 | 0.46 | 0.500 |
> | GaussianShader        | 10.48 | 0.32 | 0.620 |
> | Ours + GaussianShader | 11.97 | 0.38 | 0.622 |
>
>
> Another aspect we want to mention is that, we are *NOT* trying to directly model the translucent objects themselves, but to **perceive the Gaussian blobs that 3DGS-based methods use as absorbing bodies**. Therefore the model, i.e. each Gaussian blob, should follow the Bouguer-Beer-Lambert law.
>
> [1] Attal, Benjamin, et al. "Flash Cache: Reducing Bias in Radiance Cache Based Inverse Rendering." European Conference on Computer Vision. Springer, Cham, 2025.
>
> [2] Liang, Zhihao, et al. "Gs-ir: 3d gaussian splatting for inverse rendering." Proceedings of the IEEE/CVF Conference on Computer Vision and Pattern Recognition. 2024.
>
> [3] Jiang, Yingwenqi, et al. "Gaussianshader: 3d gaussian splatting with shading functions for reflective surfaces." Proceedings of the IEEE/CVF Conference on Computer Vision and Pattern Recognition. 2024.

---

> > ### Comment · Reviewer_RDFN · 2024-11-21
> >
> > I acknowledge that 3D-GS excels at real-time rendering. However, I do not agree that 3D-GS-based inverse rendering methods generally outperform NeRF-based methods. For instance, in Figure 5, the surface normals of the glossy ball and the toaster are nearly perfectly reconstructed by NeRF-based methods [1, 2]. In contrast, 3D-GS-based methods show obvious concave artifacts on the surface.
> >
> > Regarding Question 4, which I believe is the major issue of the work, the main motivation seems to be applicable primarily to translucent materials where opacity depends on the material properties. However, all the experiments are conducted on opaque materials. Based on the experimental results on Dex-NeRF dataset, the original 3D-GS performs much better.
> >
> > I have a question: why is the original 3D-GS considered the upper bound? I believe that with proper modeling, such as employing the Bouguer-Beer-Lambert law, your method should be able to outperform 3D-GS. For example, Ref-NeRF [3] models specular light more accurately, which leads to results that surpass the original NeRF.
> >
> > Would it be possible to apply your method to the original 3D-GS framework to further improve performance? This could potentially address the limitations highlighted in your experiments.
> >
> > [1] Liu Y, Wang P, Lin C, et al. Nero: Neural geometry and brdf reconstruction of reflective objects from multiview images[J]. ACM Transactions on Graphics (TOG), 2023, 42(4): 1-22.
> > [2] Liang R, Chen H, Li C, et al. Envidr: Implicit differentiable renderer with neural environment lighting[C]//Proceedings of the IEEE/CVF International Conference on Computer Vision. 2023: 79-89.
> > [3] Verbin D, Hedman P, Mildenhall B, et al. Ref-nerf: Structured view-dependent appearance for neural radiance fields[C]//2022 IEEE/CVF Conference on Computer Vision and Pattern Recognition (CVPR). IEEE, 2022: 5481-5490.

---

> > > ### Author Response · Authors · 2024-11-23
> > > **Replying to response from Reviewer RDFN**
> > >
> > > > 1. I do not agree that 3D-GS-based inverse rendering methods generally outperform NeRF-based methods
> > >
> > > We agree with this statement and based on this we changed the abstract accordingly (Ln. 15 in the manuscript). However, as you mentioned, 3DGS-based methods do have their pros (e.g. faster rendering, compact representation, etc.) w.r.t NeRF-based methods, therefore in this work we focus on this branch of work. In the meantime, it is applicable to also apply our method in NeRF-based frameworks. However, it requires some careful design on the sampling strategy since volume density has to be used for informing material samples which in turn has to serve as a factor that affects volume density. Therefore we respectfully argue that implementing our method into NeRF-based methods is not a straightforward thing that can be done in one week and we’d like to leave it as an interesting future direction.
> > >
> > > > 2. The main motivation seems to be applicable primarily to translucent materials where opacity depends on the material properties. However, all the experiments are conducted on opaque materials. Based on the experimental results on Dex-NeRF dataset, the original 3D-GS performs much better.
> > >
> > > We argue that the motivation of our work is not to model transparent materials but to perceive each Gaussian blob as a transparent body that can absorb light in a physics-informed way. As for the experimental results, it is not surprising that original 3D-GS performs better than inverse rendering methods. This is due to the **wrong** modeling of using BRDF for transparent objects a set of parameters that fails to model refraction, the core characteristic distinguishing transparent materials from opaque ones. The correct way of modeling it requires BSSRDF parameters, which is not included in any of the baselines and remains an open research problem.
> > >
> > >
> > >
> > >
> > > > 3. Why is the original 3D-GS considered the upper bound? I believe that with proper modeling, such as employing the Bouguer-Beer-Lambert law, your method should be able to outperform 3D-GS
> > >
> > > Since the use of BRDF parameters for transparent objects is wrong, original 3D-GS, which implicitly learns these attributes by supervision on novel view synthesis, will undoubtedly do better than a wrong physical modeling. On the other hand, the original 3D-GS serves as a reference here in our case, showing how a general model should perform approximately.
> > >
> > > The baseline that we built upon, GaussianShader[1], indeed models the reflection and shows better performance on opaque objects than the original 3D-GS. In our work, we further improve upon it, thus beating the original 3D-GS on opaque objects as well.
> > >
> > >
> > >
> > >
> > > > 4. Would it be possible to apply your method to the original 3D-GS framework to further improve performance? This could potentially address the limitations highlighted in your experiments.
> > >
> > > Thank you for this suggestion. Per your comments, we straightforwardly apply the activation function for alpha in the original 3D-GS framework and introduce an MLP that takes as input the SH coefficients and output cross section. The result is reported below.
> > >
> > > | Method                | PSNR  | SSIM | LPIPS |
> > > | --------------------- | ----- | ---- | ----- |
> > > | 3DGS                  | 17.47 | 0.46 | 0.500 |
> > > | 3DGS +Ours        | 17.54 | 0.47 | 0.501 |
> > > | GaussianShader        | 10.48 | 0.32 | 0.620 |
> > > | Ours + GaussianShader | 11.97 | 0.38 | 0.622 |
> > >
> > >
> > > As can be seen from the table, our method indeed makes the original 3D-GS better. However, since the material properties are missing in the original 3D-GS setting, the design of using SH coefficients for it may not be the best choice. We’d like to point out that our method is designed for inverse rendering and variation towards regular rendering is indeed an interesting future direction.
> > >
> > >
> > >
> > > [1] Jiang, Yingwenqi, et al. "Gaussianshader: 3d gaussian splatting with shading functions for reflective surfaces." Proceedings of the IEEE/CVF Conference on Computer Vision and Pattern Recognition. 2024.gm

---

> ### Author Response · Authors · 2024-11-27
> **We sincerely appreciate your time and valuable feedback**
>
> Dear Reviewer RDFN,
>
> We sincerely appreciate the time and effort you have devoted to reviewing our manuscript and providing valuable feedback. As the deadline for uploading visualizations and revising the manuscript is November 27th, we kindly ask if there are any remaining questions or if further clarifications are needed. We would be more than happy to provide any additional information or address any concerns.
>
> Thank you once again for your thorough review and constructive input.
>
> Best Regards,
>
> Authors

---

### Official Review · Reviewer_rqsv · 2024-11-02

**Soundness:** 3
**Presentation:** 3
**Contribution:** 4
**Rating:** 8
**Confidence:** 4

**Summary:**

The paper draws inspiration from the Bouguer-Beer-Lambert law in radiative transfer theory, and applies this theory to 3D Gaussian Splatting. The paper analyzes the alpha blending in 3DGS, considering it as an absorbing body. Based on this analysis, the paper proposes to add a material-affected term into the computation of $\alpha$ value, namely the cross section modeling, predicted by an MLP from the material parameters. The paper's design can be plugged into exisiting 3DGS-based inverse rendering pipelines, and experiments show that it can achieve superior performance compared with the original inverse rendering method.

**Strengths:**

1. Good paper writing and analysis. The paper explains its motivation and idea very clearly and intuitively, with mathematical derivations. I like the analysis subsection, which makes the paper's design even more reasonable.
2. Technical contribution and novelty. The paper points out the limitation of existing methods that disentangle opacity from material, and develops a method to add the dependency between opacity and material. I think this is a very meaningful and valuable point of view.
3. The paper experiments on 3 baseline methods in 3 different datasets, demonstrating the proposed method's applicability in different scenarios.

**Weaknesses:**

1. Some designs of the paper requires further explanation. See the "Questions" section for my confusion.
2. Citations. Some suggestions of additional citations:
    - Huang, Binbin, et al. "2d gaussian splatting for geometrically accurate radiance fields." ACM SIGGRAPH 2024 Conference Papers. 2024
    - Yariv, Lior, et al. "Volume rendering of neural implicit surfaces." Advances in Neural Information Processing Systems 34 (2021): 4805-4815.
    - Miller, Bailey, et al. "Objects as volumes: A stochastic geometry view of opaque solids." Proceedings of the IEEE/CVF Conference on Computer Vision and Pattern Recognition. 2024.
    - Munkberg, Jacob, et al. "Extracting triangular 3d models, materials, and lighting from images." Proceedings of the IEEE/CVF Conference on Computer Vision and Pattern Recognition. 2022.
3. Subtle improvement in Fig. 4. The comparison with R3DG on Synthetic4Relight dataset demonstrates little difference. I wonder if the authors can highlight some areas with more significant differences in the figure.
4. Needs more geomentry comparison. The proposed cross section modeling provides a more physical constraint on the opacity, which I think can improve the constructed geometry. The experiment section has demonstrated qualitative results on normal maps, and I think the paper should include more quantitative comparisons on the reconstructed geometry, including normal accuaracy, and the extracted mesh if possible.
5. Lack of interpretability in the cross section modeling. The paper simply uses an MLP to predict $\sigma_\nu$, which is an implicit mapping and lacks interpretability. Since the intuition of cross section modeling is based on the law of physics, the design should also be more reasonable.
6. A minor suggestion: It will be better to use vector graphics (e.g. PDF or SVG) instead of bitmaps (e.g. PNG or JPG) in the figures.

**Questions:**

1. Notation for Guassian covariance matrix. What does the $R_\theta$ mean in the Gaussian covariance matrix expression $R_\theta(\Sigma_i)^{-1}$ in Eq. 2 and 9?
2. L275-277. The explanation on why the path length $s$ can be simply set to constant 1 is not clear and convincing. The authors should provide a more detailed explanation on it.
3. Implementation details of the MLP. Does it use positional encoding for the material properties, or just directly feed the properties into the MLP?
4. Experimental results show improvements on normal estimation. Will the proposed method work better on surface-based Gaussian Splatting methods, e.g. 2D Gaussian Splatting, which has a more well-defined normal and surface geometry representation?

---

> ### Author Response · Authors · 2024-11-21
> **Response to Reviewer rqsv (Part 1/2)**
>
> Thank you for dedicating your time and providing such perceptive feedback. Your recommendations have considerably enhanced our work. Based on your comments, we proposed a plug-and-play module that perceives the alpha-blending in 3DGS as absorbing bodies. Our method archives superior performances compared to existing approaches. We will address your concerns below.
>
> > 1. Relevant citations
>
> Thank you for your suggestion. We’ve already added these relevant literatures (Section 2) into our work to make it more comprehensive.
>
> > 2. Subtle improvement in Fig. 4
>
> Thank you for carefully reading our manuscript and point this out! We agree that Fig.4 may not be the best representative of the qualitative experiments with R3DG on Synthetic4Relight dataset, therefore we added a new set of images in the appendix in Fig. C4. Please let us know if you have further suggestions!
>
> > 3. Needs more geomentry comparison. ... I think the paper should include more quantitative comparisons on the reconstructed geometry, including normal accuaracy, and the extracted mesh if possible.
>
> Thank you for your insightful comments. We carefully investigated the ground truth provided by the four datasets that we followed the baselines to conduct experiments on. We found that when synthesizing the ground truth images in simulated environments (i.e. Blender), the normal of the object is decided by the normal of the triangle on the mesh, therefore a ground truth normal map is not needed for providing ground truth images for the datasets. In that sense, the quantitative results on normal accuracy are not accessible. As for extracting mesh, we found a paper called SuGar[1] which aims to provide an algorithm for extracting mesh from 3DGS-based methods and is far beyond the scope of this paper. We agree that more understanding on the reconstructed geometry is beneficial, therefore we add more qualitative results in Fig. C5 in the Appendix.
>
> >4. Lack of interpretability in the cross section modeling.
>
> Thank you for noticing the novelty of our approach! To the best of our knowledge, there is no closed-loop form for cross section in physics. And also there is no explicit equation for mapping the types of particles to macro appearances. These are the reasons why we use a general function approximator, i.e. an MLP, for cross section modeling.
>
> >5. It will be better to use vector graphics
>
> Thank you for carefully investigating our manuscript! We’ve already changed the figures in the paper to SVG format per your suggestions!
>
> >6. Notation for Guassian covariance matrix.
>
> Thank you for your question! The notation means the projection based on camera parameters applied on the covariance matrix. The expression follows 3DGS-MCMC [2] and we’ve added some clarification in Ln 185-186 in the text to make it clear.

---

> ### Author Response · Authors · 2024-11-21
> **Response to Reviewer rqsv (Part 2/2)**
>
> > 7. The explanation on why the path length $s$ can be simply set to constant 1 is not clear and convincing. The authors should provide a more detailed explanation on it.
>
> Thank you for pointing this out! We modified the manuscript in Ln. 265 - Ln. 269 to add an explanation to this design choice. In general, since all the 3D Gaussians are projected to the 2D plane, the blending algorithm omits the depth of the Gaussians into the formulation of the opacity value at the pixel location. We therefore observed that we can treat this opacity value as the density and since there is no concept of depth on the 2D plane, each Gaussian is then treated equally from this perspective, therefore it is reasonable to treat the path length to constant 1.
>
> >8. Implementation details of the MLP. Does it use positional encoding for the material properties, or just directly feed the properties into the MLP?
>
> Thank you for noticing this! We added the implementation details of the MLP in the Appendix from Ln. 768 - Ln. 770. In one word, we directly feed the material properties into the MLP without any positional encoding.
>
> >9. Will the proposed method work better on surface-based Gaussian Splatting methods, e.g. 2D Gaussian Splatting, which has a more well-defined normal and surface geometry representation?
>
> Thank you for providing this insightful comment! Nonetheless, as no existing approach employs 2D Gaussian Splatting for inverse rendering, and our method is specifically designed for inverse rendering, directly applying it to 2D Gaussian Splatting is not straightforward.  That said, based on the observed improvements in normal estimation in our experiments, we hypothesize that our method could enhance the accuracy of normal estimation in 2DGS if applied to inverse rendering. Since it requires significant effort to design a pipeline utilizing 2DGS for inverse rendering, we’d like to retain this as an interesting future research direction.
>
> [1] Guédon, Antoine, and Vincent Lepetit. "Sugar: Surface-aligned gaussian splatting for efficient 3d mesh reconstruction and high-quality mesh rendering." Proceedings of the IEEE/CVF Conference on Computer Vision and Pattern Recognition. 2024.
>
> [2] Kheradmand, Shakiba, et al. "3D Gaussian Splatting as Markov Chain Monte Carlo." arXiv preprint arXiv:2404.09591 (2024).

---

> ### Comment · Reviewer_rqsv · 2024-11-22
>
> Thanks for your reply! Most of my concerns have been addressed, but I have a few suggestions below:
> ### Geometry comparison
> > We found that when synthesizing the ground truth images in simulated environments (i.e. Blender), the normal of the object is decided by the normal of the triangle on the mesh, therefore a ground truth normal map is not needed for providing ground truth images for the datasets.
>
> I believe that the ground truth normal map can be obtained by a simple rasterization / ray casting process from the mesh in the rendering pipeline, also known as a G-buffer (geometry buffer). I personally think this is an important comparison and strongly encourage the authors to include this.
> ### Cross section modeling
> I acknowledge that cross section modeling does not have closed-form physical formula, but I think it will be interesting to qualitatively illustrate the $\sigma_\nu$ value learned from the MLP to see if the result is consistent with the human intuition.
> ### Subtle improvements
> It will be nice if the authors can highlight the areas in the results figures that demonstrate improvements over baselines (in Fig 4 and Fig C4), maybe using arrows or insets. I have to zoom in to carefully find out the difference by now. And it will also be nice to include quantitative metrics next to the figure to further emphasize the differences.

---

> > ### Author Response · Authors · 2024-11-25
> > **Replying to Reviewer rqsv**
> >
> > > 1. Geometry comparison
> >
> > Thank you for the suggestion! Since only the Shiny Blender dataset provides the source file to produce the ground truth normal map, we evaluate the GaussianShader model as well as our model on it. The MAE metric used for normal evaluation shows that our model is slightly better than the baseline model (baseline: 32.323 v.s.ours: 32.269). While our model indeed show better qualitative results, we hypothesize that the improvement arises because our method applies regularization to the material properties and implicitly influences the optimization process to make the model produce smoother prediction (despite still not perfect). The marginal improvement in normal estimation is not surprising to see since we didn’t add any supervision or regularization to the normal attributes. We’d like to retain this interesting finding as future work to study a better modeling for normal estimation using 3DGS.
> >
> > > 2. Cross section modeling
> >
> > Thank you for your constructive comment that helps us make our work more comprehensive. We follow your suggestion to visualize the trained point cloud on the Mip-NeRF 360 dataset as it contains more diverse material. We assigned the color of each point to correspond to the cross section. We use the jet color map to visualize it in MeshLab. We do find some results that correspond to human intuition. Some observations are as follows: for the bonsai scene in the Mip-NeRF 360 dataset, the plastic bag on the ground is assigned the lowest cross section in the whole point set (meaning it lets more light to pass through), the window on the far end of the counter scene has low cross section as well. We are working on better visualizations and will include the results in the final version of the paper.
> >
> > > 3. Subtle improvements
> >
> > Thank you for the constructive comment! We agree it’s difficult to tell the visual difference so we added some zoom-in views to Fig. 4 and Fig. C4 to make it more clear. Please let us know if it is sufficient to tell the difference! For these figures, we highlight the differences on albedo, roughness and the resulting rendering and relighting results. For instance, for Fig. 4, the string on the jugs shows incorrect albedo for the baseline method (a little bit red) while our prediction is closer to the ground truth. As for the roughness, check out also in Fig. 4 that our method predicts a more reflective surface inside the jugs. As a result, the rendering of our method in Fig. 4 shows reflection closer to ground truth on the surface of the jugs and shows correct color at the bottom of the jugs when relighting. Fig. C4 is also highlighted in this way, where our method provides smoother and correct roughness compared to baseline and also correct color in albedo (also on the string). As a result the rendering and relighting also demonstrate the direct effect of such correct prediction.

---

> > ### Author Response · Authors · 2024-11-25
> > **Replying to Reviewer rqsv on cross section visualization**
> >
> > Thank you again for the constructive comments on visualizing the cross section to make our work more comprehensive. We added some rendered 2D maps of the cross section in Fig. D6 and Sec. D in the Appendix. In the figures, darker regions are with low cross section, meaning that they let more light pass through, brighter regions are vice versa. The visualization results do show some intuitive effects. For example, windows and plastic bags have low cross section (dark regions pointed by the arrows), meaning that they let more light pass through, which corresponds to human intuition.

---

> > > ### Comment · Reviewer_rqsv · 2024-11-26
> > >
> > > Thanks for your reply and revision. I'm satisfied with the results provided, and I'm considering increasing the rating to Accept. However, I still have several minor suggestions:
> > > ### Subtle Improvements
> > > I currently find the qualitative relighting results for the Synthetic4Relight dataset too dark to recognize the details. I suggest increasing the exposure level during the rendering for a better visualization.
> > > ### Cross Section Visualization
> > > I appreciate the updated results, and agree that they correspond to human intuition. A suggestion is that the authors can visualize the cross-section results together with the material parameters, since materials are the input of the cross-section network, so that we can see the correspondence. Also, I wonder whether the cross-section parameters should be visualized in a color map or in black and white (current version).
> > > ### Discussion about Limitation
> > > From the cross-section visualization, I find the marble table top in the Counter scene contains artifacts. I think this is because of the specularity, and the authors should add an analysis paragraph of it as a limitation and a possible future work of this paper.

---

> > > > ### Author Response · Authors · 2024-11-26
> > > > **Replying to Reviewer rqsv on visualization suggestions**
> > > >
> > > > Thank you for your detailed suggestions to help us deliver our work better!
> > > >
> > > > We've modified the manuscript to include brighter relighting results in Fig. 4 and Fig. C4.
> > > >
> > > > Also, Fig. D6 is modified to include material properties and rendering results for correspondence.
> > > >
> > > > Last but not least, the limitation section is expanded to discuss the visualization results in Fig. D6 to motivate future works on learning a better model for specular objects and surfaces.
> > > >
> > > > Please let us know if you have further suggestions to help us improve the presentation!

---

> > > > > ### Comment · Reviewer_rqsv · 2024-11-26
> > > > >
> > > > > Dear authors,
> > > > > Thanks for your revision and reply! Most of my questions and concerns have been addressed, and the manuscript has become more informative after revisions. I'll keep Accept as my review recommendation.

---

> > > > > > ### Author Response · Authors · 2024-11-26
> > > > > > **Grateful for Your Recognition and Positive Recommendation**
> > > > > >
> > > > > > We greatly appreciate your recognition of our work's novelty and contribution. We sincerely thank you for your thoughtful feedback and are grateful for your positive recommendation.

---

### Official Review · Reviewer_G4SH · 2024-11-03

**Soundness:** 2
**Presentation:** 3
**Contribution:** 2
**Rating:** 6
**Confidence:** 5

**Summary:**

This paper proposes to model opacity, as derived from the existing material properties, to improve Gaussian Splatting for better rendering. The work is inspired by the physical phenomenon that light intensity decays at different rates as it travels through different absorbing media. The authors map existing terms in the  GS formulation to terms in the Bouguer-Beer-Lambert law and add a new term predicted from albedo, metalness, and roughness, to cover the additional cross-section term.

The authors also provide nice interpretations of their formulation from other perspectives such as Nerf and Taylor expansion, thereby stressing the correctness of their approach.

Since the method is plug-and-play, it was run on synthetic datasets containing glossy objects and also on real-world datasets such as Mip-nerf 360. Both the qualitative and quantitative results show that the method achieves high visual quality.

**Strengths:**

I like how this work incorporates a physically-based factor into the existing GS formulation. The additional term required to cover cross section also makes sense, since that should indeed be material-dependent and disentangled from geometry.

I also like the interpretations the authors provide, to view their proposed approach from other perspectives. That facilitated understanding and also boosted the reader’s confidence to a certain degree.

The method is plug and play, so if working well, it can have a big impact on how people model Gaussian Splats.

**Weaknesses:**

It’s surprising that this paper shows no result of transparent/translucent objects. I don’t think it suffices for a method designed to handle transparent/tanslucent appearance to show improvements on opaque objects but not on the original objects of interest. Ideally, there should be videos of view synthesis results on translucent objects as evidence to what this paper claims.


Assuming this method does handle translucency well, the big question mark in my head then is how the model deals with shape-material ambiguity. As a specific example: a cup could be due to regular cup geometry paired with regular cup material or cube geometry paired with a translucent material that’s transparent everywhere and opaque around/in the center cup shape. There should be some analysis on why this is not happening if the paper properly models transparency/translucency. A specific experiment the authors can do is setting up a synthetic scene where a shape is enclosed by some translucent material. Does the model reconstruct the translucent part or just ignore it, reconstructing the object directly?

Also, current improvements reported in the paper on opaque objects may simply be due to a bigger model / gaussian formulation with more parameters we can optimize. If this is actually the case, the whole BBL law inspiration no longer stands.

I am not fully convinced that the cross section network is really learning cross section from albedo, metalness, and roughness. In fact, I don’t think that makes sense.  Consider a translucent material that has some cross section parameter for the network to learn. This material may not have well-defined albedo, metalness, or roughness, which are properties in opaque appearance BRDFs. I hope the authors can provide some clarifications around this (see also my questions below).

**Questions:**

Related to the biggest weakness IMO above, have the authors tried this on translucent objects? If this doesn’t work on those objects, the basis of this paper is fundamentally challenged.

Related to my final point in Weaknesses, have the authors tried to understand what the cross section network is learning? This can be done in many ways, but the one I have in mind is seeing how a trained network responds to varying inputs of roughness, metalness, and albedo. Again, I’m very skeptical of the network learning anything physically-based. Why would albedo inform cross section? Again, they seem to be from two different worlds of material properties.

Nit: I like Figure 1a, but Figure 1b is misleading until I got to the very end of the paper, where the authors said their approach didn’t model the dependency on light frequency. I recommend removing Figure 1b if color of the light doesn’t matter in the entire paper.

---

> ### Author Response · Authors · 2024-11-21
> **Response to Reviewer G4SH (Part 1/2)**
>
> We’re thankful for your time and the valuable insights you’ve shared. Your input has significantly advanced our project. In response to your feedback, we proposed to model opacity based on the Bouguer-Beer-Lambert law by introducing a cross-section term and it achieved superior performance across various datasets. We’ll address your concerns below.
>
> > 1. No result of transparent/translucent objects.
>
> Per your comments, we conducted experiments on the Dex-NeRF dataset, the existing translucent object dataset for 3D reconstruction. Notice that there does not exist any inverse rendering baseline for translucent objects. To make a comparison, we run experiments using the GaussianShader as baseline and ours + GaussianShader as the demonstration of the effectiveness of our method. As can be seen from the results, our method still outperformed the baseline method. However, since albedo, roughness and metalness are not the parameters for describing translucent objects, we argue that developing such a method is out of the scope of this work.
>
> | Method                | PSNR  | SSIM | LPIPS |
> | --------------------- | ----- | ---- | ----- |
> | 3DGS                  | 17.47 | 0.46 | 0.500 |
> | GaussianShader        | 10.48 | 0.32 | 0.620 |
> | Ours + GaussianShader | 11.97 | 0.38 | 0.622 |
>
>
> On the other hand, we’d like to point out that our motivation is *NOT* based on modeling translucent objects, but on **each Gaussian point is itself a translucent body that has density and cross section to describe its absorption coefficient, namely opacity**. So the whole Gaussian Splatting pipeline is to use a set of translucent bodies to model objects, no matter if it's translucent or opaque. Therefore, the modeling itself, i.e. each Gaussian blob, should follow the Bouguer-Beer-Lambert law per our design.
>
> > 2. Current improvements reported in the paper on opaque objects may simply be due to a bigger model with more parameters we can optimize
>
> Thanks for your valuable suggestion. Parameters of a 3DGS model mainly come from the large number of points and each point has 24 parameters in the inverse rendering setting (take GaussianShader here as an example). We observe that our model on average (for GaussianShader on GlossySynthetic dataset) only has 3.29E+5 points and the original has 3.44E+5 points. Consider the MLP we have (Added in the Appendix on the architecture), which in total only contains 17,665 parameters (approximately 0.2% of the total number of parameters), our model’s parameters are even less than the baseline model. As for different datasets, the number of points is reported in the table. We therefore argue that the performance improvement does not come from increasing parameters but from our design.
>
> | Datasets | GlossySynthetic | Shiny Blender | Mip-NeRF 360 | Synthetic4Relight |
> | -------- | --------------- | ------------- | ------------ | ----------------- |
> | Baseline | 3.44E+05        | 1.84E+05      | 4.38E+06     | 2.40E+05          |
> | Ours     | 3.29E+05        | 1.83E+05      | 3.87E+06     | 2.09E+05          |

---

> ### Author Response · Authors · 2024-11-21
> **Response to Reviewer G4SH (Part 2/2)**
>
> > 3. I am not fully convinced that the cross section network is really learning cross section from albedo, metalness, and roughness. ... Consider a translucent material that has some cross section parameter for the network to learn. This material may not have well-defined albedo, metalness, or roughness, which are properties in opaque appearance BRDFs.
> ...  Why would albedo inform cross section?
>
> Thank you for pointing this out. We’d like to argue that based on our experimental results that show no parameter increasement, we found that the performance improvements didn’t come from more parameters. Therefore, our cross-section network effectively captures meaningful dynamics, enabled by the thoughtful modeling we introduce.
>
> On the other hand, as you mentioned in your statement, there doesn’t exist any methods that do inverse rendering for translucent objects and it’s also not the focus of our work, such a topic is beyond the scope of this paper.
>
> The reason albedo can inform cross section can be intuitively understood as follows: in an absorbing body, the cross section is influenced by the type of the particles it contains, making the modeling of particle types the key challenge. On the other hand, the type of the particles can be represented by the albedo, roughness and metalness of the body because on the macro level different types of particles appear differently and can be described by different appearances.
>
> > 4. Figure 1b is misleading
>
> Thank you for pointing this out. We agree that Figure 1 is a little misleading. The point of using these figures is that we want to demonstrate that no matter how the light changes, different materials would react differently. We’d like to point out again that our motivation comes from the observation that each Gaussian blob is an absorbing, translucent body, instead of directly modeling translucent objects themselves. We’ve changed Figure 1 per your suggestion. Please let us know if you find it still inappropriate.

---

> ### Author Response · Authors · 2024-11-27
> **We truly value your time and thoughtful feedback**
>
> Dear Reviewer G4SH,
>
> Thank you for the time and effort you have dedicated to reviewing our manuscript and providing valuable feedback. As the deadline for submitting visualizations and revised manuscripts approaches on November 27th, we wanted to check if there are any outstanding questions or if further clarification is needed. We would be glad to provide any additional information as required.
> We deeply appreciate your insights and look forward to your guidance.
>
> Best regards,
>
> Authors

---

> > ### Comment · Reviewer_G4SH · 2024-11-29
> > **Helpful clarifications**
> >
> > Thank you for the clarifications, especially the explanation that the goal of this paper is to use translucent blobs to represent mostly opaque scenes. This is key, and based on that, I'm willing to improve my score to marginally above threshold.

---

### Official Review · Reviewer_gEKk · 2024-11-04

**Soundness:** 4
**Presentation:** 4
**Contribution:** 3
**Rating:** 8
**Confidence:** 3

**Summary:**

This paper introduces a plug-and-play approach for enhancing Gaussian Splatting in material modeling by explicitly modeling the relationship between opacity and material properties using the Bouguer-Beer-Lambert law. The authors validate this insight through various analyses, demonstrating its correctness. Qualitative results across different methods and datasets show consistent improvements, as evidenced by performance gains across multiple metrics.

**Strengths:**

1. The core idea is pretty simple and straightforward, which I personally enjoyed. The authors have demonstrated the validity of their formulation from multiple perspectives, making the approach theoretically sound and insightful.
2. The proposed formulation leads to improved reconstruction quality across various datasets and methods.
3. This paper is well written and easy to follow. The entire formulation is physically sound and all the theoretical details are explained in an intuitive way for readers.
4. Given the simplicity of this method, it should be broadly applicable to future 3DGS-based material modeling approaches, potentially making a significant impact to the community.

**Weaknesses:**

1. While the reported quantitative metrics indicate improvement, I struggled to notice significant visual differences in the rendered RGB images. The geometric variations are more apparent in some normal renderings. Maybe additional results or visualizations could be provided to better illustrate the quality difference.
2. As mentioned by the authors, the impact of lighting frequency/spectrum is not considered, which could be an interesting future direction to model more complex materials.

**Questions:**

Given the simplicity of this formulation and the well-written manuscript, I don't have further questions regarding this submission. In general I enjoy reading this paper and love this simple yet effective solution. Just look at some minor issues in Weaknesses.

---

> ### Author Response · Authors · 2024-11-21
> **Response to Reviewer gEKk**
>
> We are grateful for your time and insightful comments. Your valuable suggestions have significantly elevated our work. In light of your comments, we introduce a simple yet effective plug-and-play approach for inverse rendering using Gaussian Splatting and demonstrate its usefulness through various analyses and experiments. We will address your concerns below.
>
> > 1. Additional results or visualizations could be provided to better illustrate the quality difference.
>
> Thanks for your comment. We also notice that the visualization result of Synthetic4Religiht dataset does not make the effectiveness of our method clear. Therefore we chose another set of images and updated the manuscript in Appendix Fig. C4. For the other 2 visualization results, MIP-NeRF 360 is made clear with the zoom-in window (artifacts on the bicycle and on the tablecloth) while for the glossy datasets, the main intuition is to showcase the superior performance of our method in terms of better normal estimation. Please let us know if you find any of them unsatisfactory.
>
> > 2.  As mentioned by the authors, the impact of lighting frequency/spectrum is not considered, which could be an interesting future direction to model more complex materials.
>
> Thank you for pointing this out.  Since the purpose of this work is to introduce the concept of the Bouguer-Beer-Lambert law into the field, we plan to introduce lighting frequency as future work, i.e. designing a more sophisticated and accurate model for the relation between opacity and materials. This is also mentioned in the Limitations section in our manuscript.

---

> ### Comment · Reviewer_gEKk · 2024-11-23
>
> Dear authors,
>
> Thanks for your reply! Most of my concerns are addressed. My remaining question is about the visual comparison. As pointed by other reviewers (Reviewer rqsv), both Figure 4 and Figure C4 include visuals with subtal differences, which are hard to distinguish. Maybe a color visualizaion of MSE or zoomed-in insets could be provided to highlight the difference between the proposed method and baselines.

---

> > ### Author Response · Authors · 2024-11-25
> > **Replying to Reviewer gEKk**
> >
> > > Subtle improvements
> >
> > Thank you again for your positive feedback and the valuable suggestion! We added some zoom-in views in Fig. 4 and Fig. C4 to demonstrate the visual difference per your comment, please let us know if it is clear enough! For these figures, we highlight the differences on albedo, roughness and the resulting rendering and relighting results. For instance, for Fig. 4, the string on the jugs shows incorrect albedo for the baseline method (a little bit red) while our prediction is closer to the ground truth. As for the roughness, check out also in Fig. 4 that our method predicts a more reflective surface inside the jugs. As a result, the rendering of our method in Fig. 4 shows reflection closer to ground truth on the surface of the jugs and shows correct color at the bottom of the jugs when relighting. Fig. C4 is also highlighted in this way, where our method provides smoother and correct roughness compared to baseline and also correct color in albedo (also on the string). As a result the rendering and relighting also demonstrate the direct effect of such correct prediction.

---

> > > ### Comment · Reviewer_gEKk · 2024-11-25
> > >
> > > Dear authors,
> > >
> > > Thanks for your update! These figures look good to me and I tentatively have no further questions!

---

> > > > ### Author Response · Authors · 2024-11-26
> > > > **Thank You for Your Encouragement and Recognition**
> > > >
> > > > We are truly encouraged by your recognition of the novelty of our work. Thank you once again for your insightful feedback, we greatly appreciate your positive recommendation!

---

### Author Response · Authors · 2024-11-21
**Response to all reviewers**

# General Response
We’d like to thank all the reviewers for their valuable feedback, especially for acknowledging our contributions regarding the novel design based on the physical law and the effort we made to validate and show the plug-and-play attribute and effectiveness of our method. Regarding common concerns among reviewers, we have added some experiments and present the results to address these concerns.

----
## Clarification on the motivation
Since the Fig.1 in our manuscript has some misleading effect on the readers, we’ve already modified the manuscript and would like to make a clarification here on the motivation of the paper. We propose to perceive a 3DGS-based model as a set of translucent bodies and view the alpha-blending algorithm as radiative transfer described by the Bouguer-Beer-Lambert law. Therefore we introduce a material-dependent factor, namely cross section, to each of the translucent (absorbing) bodies, i.e. each Gaussian blob. Based on these observations and design choices, we derived the whole algorithm discussed in the paper. In summary, the goal of the paper is not to propose a way for inverse rendering of transparent objects, but to propose a physically informed modeling of the term opacity that prevails in the 3DGS-based methods.

----
## Transparent / Translucent objects
While there is no existing inverse rendering method that targets at modeling transparent / translucent objects, we directly apply one of the baseline methods in our paper, namely GaussianShader, to the Dex-NeRF dataset. We use the real-world subset of the dataset. Since it originally doesn’t have a train-test split, we randomly select $\frac{2}{3}$ of the entire image set for each scene to be the training set and the rest to be the test set. We also run the original 3DGS on this dataset to demonstrate an “oracle” performance an inverse rendering model should have. As can be seen from the results, because the BRDF parameters are not designed for transparent / translucent objects, the inverse rendering method fails to provide convincing results. Despite all that, our method still slightly outperformed the baseline due to the regularization we add to the modeling of opacity. We argue that developing an entire new set of parameters for transparent / translucent objects remains an open research question and is beyond the scope of this paper.



| Method                | PSNR  | SSIM | LPIPS |
| --------------------- | ----- | ---- | ----- |
| 3DGS                  | 17.47 | 0.46 | 0.500 |
| GaussianShader        | 10.48 | 0.32 | 0.620 |
| Ours + GaussianShader | 11.97 | 0.38 | 0.622 |

---

### Comment · Area_Chair_MNt8 · 2024-11-25
**Please read the rebuttal and reply**

Dear Reviewers,

Thanks again for serving for ICLR, the discussion period between authors and reviewers is approaching (November 27 at 11:59pm AoE), please read the rebuttal and ask questions if you have any. Your timely response is important and highly appreciated.

Thanks,

AC

---

### Meta-Review · Area_Chair_MNt8 · 2024-12-20

**Metareview:**

This paper proposes a plug-and-play method to enhance 3DGS for inverse rendering. Specifically, it proposes to associate opacity prediction with material following the Bouguer-Beer-Lambert law. This paper is well written and easy to read. The proposed method is simple and intuitively reasonable.

During rebuttal, reviewers raised questions about performance verification, method effectiveness, missing citations, most of these concerns are addressed during the rebuttal. All reviewers agree to accept the paper, thus the paper is recommended for accpetance.

**Additional Comments On Reviewer Discussion:**

During rebuttal, reviewers raised questions about:
- quality verification (Reviewer gEKk, rqsv, RDFN)
- missing results on transparent objects (Reviewer G4SH, RDFN)
- validation of the cross section network (Reviewer G4SH, rqsv)
- missing citations (Reviewer rqsv)

Most of these concerns are addressed during the rebuttal.

---

### Decision · Program_Chairs · 2025-01-22

Accept (Poster)